# Evolutionary Kuramoto dynamics unravels origins of chimera states in neural populations

Thomas Zdyrski[1]*, Scott Pauls[1], Feng Fu[1,2]

**1** Department of Mathematics, Dartmouth College, Hanover, New Hampshire, United States of America, **2** Department of Biomedical Data Science, Geisel School of Medicine at Dartmouth, Hanover, New Hampshire, United States of America

\* thomas.zdyrski@dartmouth.edu

## Abstract

Neural synchronization is central to cognition. However, incomplete synchronization often produces chimera states, where coherent and incoherent dynamics coexist. Recent studies have suggested that these chimera states could be important in human cognitive organization. In particular, chimera states have been suggested as a regulator of cognitive integration and regulation with varying quality as humans age. While previous studies have explored such chimera states using networks of coupled oscillators, it remains unclear why neurons commit to communication or how chimera states persist. Here, we investigate the coevolution of neuronal phases and communication strategies on directed, weighted networks where interaction payoffs depend on phase alignment and may be asymmetric due to unilateral communication. The graph structure enables us to apply a game-theoretic model of Kuramoto-like oscillators to brain connectomes, and the asymmetry captures biochemical differences between communicative and non-communicative neurons. Combined, these two generalizations enable us to apply the computationally-tractable game-theoretic model of Kuramoto models to realistic brain networks and analyze the role of connectome structure on neuron communication. We find that both connection weights and directionality influence the stability of communicative strategies—and, consequently, full synchronization—as well as the strategic nature of neuronal interactions. Applying our framework to the *C. elegans* connectome, we show that emergent payoff structures, such as the staghunt game, control population dynamics. We demonstrate that weighted, directed connectivity in the *Caenorhabditis elegans* (*C. elegans*) connectome is sufficient to generate robust chimera states modulated by payoff asymmetries. Our computational results demonstrate a promising neurogame-theoretic perspective, leveraging evolutionary graph theory to shed light on mechanisms of neuronal coordination beyond classical synchronization models.

**Data availability statement:** The raw and processed data for all simulated parameter ranges is available on Zenodo at https://doi.org/10.5281/zenodo.19235754. An interactive webpage for exploring the data set is available at https://tzdyrski.github.io/egt-kuramoto/notebooks/EKT_Plots.html. Additionally, all source code is available via GitHub at https://github.com/TZdyrski/egt-kuramoto/tree/1.1.1 or Zenodo at https://doi.org/10.5281/zenodo.19582421.

**Funding:** FF received funding through the Dartmouth Scholarly Innovation & Advancement Awards (SIAA). The funders had no role in study design, data collection and analysis, decision to publish, or preparation of the manuscript.

**Competing interests:** The authors have declared that no competing interests exist.

## Author summary

Brain neurons fire together in synchronized patterns, and these synchronization waves are thought to be important for understanding human cognition. Given the miniscule scale and immense number of neurons, it is challenging to perform experimental, in-vivo studies connecting individual neuron behavior to brain-scale synchronization patterns. Evolutionary game theory (EGT) is a useful tool for analyzing how the properties of individual players affect the large-scale dynamics. In this study, we analyze the 302-neuron brain of the *C. elegans* nematode using EGT by treating the neurons as players and synapse connections as games. Numerical simulations show that the structure of the *C. elegans* brain enhances brain states known as chimeras. These chimera states display both synchronization and disorder, and they are known to be crucial indicators of brain function. Therefore, this model offers a novel framework for studying these chimera states in other brain systems by connecting small-scale neuron properties to large-scale synchronization dynamics.

## 1 Introduction

Evolutionary game theory (EGT) is the application of game theory to evolving populations of individuals with behavioral strategies. This tool is useful for studying how local interaction rules yield large-scale patterns such as cooperation [1] and has found use in fields including international politics, ecology, and protein folding [2]. Studies [3–6] have even applied EGT to non-reproducing neurons by viewing neuron plasticity as an evolutionary process where firing patterns change and are "learned" over time. These studies use a neurogame-theoretic approach: players are neurons; strategies are dynamical properties like firing rates [3] or activation/inhibition [4]; and payoffs arise from firing rate stability [3] or emulative/non-emulative neuron tendencies [4].

Evolutionary *graph* theory places evolutionary games on graphs to investigate the role of structure in population evolution. Prior studies have found that structure qualitatively changes game evolution. For instance, cooperation is enhanced by small-degree nodes [7] or unidirectional edges [8] and suppressed by weighted edges [9]. Thus, the study of evolving, structured populations should account for the effects of incompleteness, directedness, and weightedness.

Kuramoto networks are groups of coupled oscillators where the coupling strength depends sinusoidally on the oscillators' phase difference. These networks are popular models for neuron behaviour [10,11] because they exhibit tunable synchronization. Prior studies [12] have modelled synchronization with Kuramoto oscillators using EGT applied to the prisoner's dilemma game type. Other studies [5] generalized this approach to include dynamically changing game types by introducing an evolutionary Kuramoto (EK) model to show how the relationship between communication benefit and cost influences the emergence of synchronized communication (*C*) or non-communication (*N*) regimes. While the application of EGT to neurons is ultimately a mathematical abstraction, it is a

useful one [3–6] with potential biophysical underpinnings. For example, the communicative state $C$ represents an active state (either bursting or spiking) while $N$ represents its inactive state [6]. Evolution, or strategy-update, then occurs through neuro-modulation enhancing or suppressing neural activation [13]. In the evolutionary Kuramoto model, the discrete phase can be understood as the true (continuous) phase's periodic measurement: then, a constant phase represents a periodic firing at the measurement frequency while a slowly varying phase represents a frequency mismatch.

One intriguing aspect of neuron oscillations is the observation of chimera states [14]. These states exhibit the simultaneous existence of coherent and disordered phases [15]. These chimera states have appeared in a variety of topologies, including ring oscillators [15], two-dimensional spatial regions [16], and structured graphs [17]. Chimera states arise when a population splits into subcommunities, with some synchronized and others disordered; these subcommunities can be grouped by spatial region (for spatial systems) or by connectivity (for graph systems). Previous studies have proposed chimera states as a key component of human cognitive organization [18], a facilitator of spiking and bursting phases [19], and a regulator between cognitive integration and segregation [20]. Despite the observed importance of these chimera states, the factors that give rise to coherent/disordered coexistence remain incompletely characterized.

The nematode *Caenorhabditis elegans* (*C. elegans*) is a model organism in neuroscience due to its simple brain connectome [21] of only 302 neurons. Despite their simplicity, models of the *Caenorhabditis elegans* (*C. elegans*) brain still display a wide array of complex phenomena including topologically-central rich clubs crucial to motor neurons [22], phenomenological connections to control theory [23], and chimera states [24].

In this paper, we introduce an asymmetric evolutionary Kuramoto model on graphs and analyze its chimera-like states on the *Caenorhabditis elegans* (*C. elegans*) connectome. The newly introduced graph structure is necessary for modeling these Kuramoto-like players on structured populations, such as neuron connectomes, and it can qualitatively change population dynamics by selectively enhancing or suppressing cooperation [25]. This represents a new extension of the neurogame-theoretic framework by combining both Kuramoto-type oscillators for representing neuron phases and graph structure for representing connectomes. Furthermore, this payoff asymmetry between communicators ($C$) and non-communicators ($N$) models the inherent asymmetry of pre- and post-synaptic junctions: if the source neuron is $N$ and the destination neuron is $C$, then the destination neuron's payoff is modulated by $2\alpha$, which represents its receptivity to other, non-local chemical neurotransmitters [26]; conversely, if the source neuron is $C$ and the destination neuron is $N$, then the destination neuron's payoff is modulated by $2(1-\alpha)$, representing its ability to spontaneously fire in absence of upstream stimuli [27]. Our results connect individual neuron fitness and non-trivial brain topology to chimera-like brain states. Given current technological limitations with direct measurement of in-vivo neuron activity, frameworks like ours create testable hypotheses connected to the theory's assumptions. Our computational model represents a simple yet versatile framework to illuminate the influence of neural connectivity on chimera-like brain states beyond classical synchronization models.

## 2 Results

### 2.1 Model

We extend the EK model in two ways: placing the player network on a directed, weighted graph; and introducing a payoff asymmetry. First, we represent a well-mixed population as a complete graph in Fig 1A with players represented by nodes and games by edges. We generalize this to directed graphs where game (edge) payoffs only flow to the head players (nodes). We can also represent bidirectional games with a pair of edges in both directions, as shown in Fig 1B. For reference, the *Caenorhabditis elegans* (*C. elegans*) connectome has 38 self-loops, 669 bidirectional edge pairs, and 2,331 unpaired edges. Finally, we interpret the *Caenorhabditis elegans* (*C. elegans*) chemical connectome's integer-valued edge weights as the number of connections between nodes, so these weights scale each payoff.

We also generalize the payoff structure to incorporate an asymmetry between communicators and non-communicators. We characterized each player (node) by its strategy $s_i$—either communicative $C$ or non-communicative $N$— and its phase $\phi_i$—taking one of $m = 20$ evenly-spaced values between 0 and $2\pi$. When exactly one partner is communicative, the

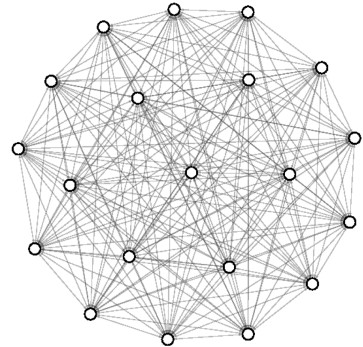
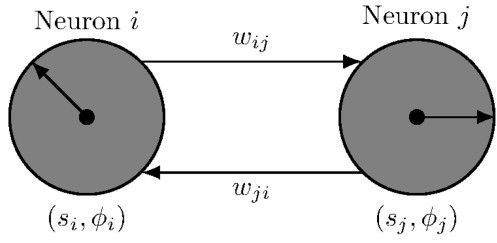

Strategy $s_i$:  **C**: communicative
**N**: non-communicative

Phase difference: $\Delta\phi = \phi_j - \phi_i$

$$
\begin{array}{c}
\\
C, \phi_i \\
N, \phi_j
\end{array}
\begin{array}{cc}
C, \phi_i & N, \phi_j \\
\left[\begin{array}{c} B_0 f(0) - c \\ 2(1-\alpha)\beta_0 f(\Delta\phi) \end{array}\right. & \left.\begin{array}{c} 2\alpha\beta_0 f(\Delta\phi) - c \\ 0 \end{array}\right]
\end{array}
$$

**Fig 1. Evolutionary Kuramoto dynamics with weighted neural connectivity. (A)** The graph of a well-mixed population with $N = 20$ players where each pair of players is connected by a directed edge in each direction. **(B)** The connectivity between two sample players, $i$ and $j$, showing directed, weighted edges $w_{ij}$ and $w_{ji}$. Each player has a strategy (communicative $C$ or non-communicative $N$) and phase $\phi = 2\pi k/m$ with $k \in 0, \ldots, m-1$ and $m$ the number of phases. **(C)** The payoff matrix shows the reward the row-player ($C, \phi_i$) receives after playing a game with the column-player ($N, \phi_j$) assuming either player can switch strategy and phase to the other's.

asymmetry $\alpha \in [0, 1]$ biases the payoff toward (against) the communicator when $\alpha$ is greater (less) than 1/2, while $\alpha = 0.5$ reproduces the symmetric case. Fig 1C shows the payoff matrix for these mixed $CN$ or $NC$ interactions and includes the maximum joint benefit $B_0$, maximum mixed benefit $\beta_0$, cost $c$, and sinusoidal Kuramoto coupling $f(\Delta\phi) = [1 + \cos(\phi_j - \phi_i)]/2$.

## 2.2 Parameter space

One of the key aspects of the EK model is that multiple 2×2 game types can emerge among the players during the population's evolution. These game types [28] include dilemma (*a.k.a.* "prisoner's dilemma"), deadlock ("anti-prisoner's dilemma"), chicken ("hawk-dove" or "snowdrift"), hero ("Bach or Stavinsky" with the lowest two payoffs swapped), harmony (game with strong incentive alignment), or concord (similar to harmony, weaker incentives). See Section 2 of the S1 Text in S1 Appendix for order graphs depicting the payoff structure of each game type. While $CC$ interactions are always double-cooperation games (denoted "cooperation" hereafter for simplicity) and $NN$ interactions are always neutral games, mixed ($CN$ or $NC$) games show a variety of game types, (Fig 2). We can visualize these games by looking at two-dimensional slices of the three-dimension phase space characterized by $\alpha$, $\beta := \beta_0 f(\Delta\phi)$, and $B_0$. Fig 2A shows a $\beta - B_0$ slice of phase space and generalizes Fig 1 of a prior study [5], while Fig 2B shows a $\beta - \alpha$ slice. Each straight line of white dots represent permissible values of the $m = 20$ phases $\Delta\phi_i$ across our simulations and highlight which regions of parameter space are accessible. The table in Fig 2C shows some select transitions as $\Delta\phi_i$ varies from 0 to $\pi$. For instance, when $B_0/c = 1.5$—corresponding to the line of white dots in Fig 2A—$\Delta\phi = 0$ starts in the chicken region and moves left, transitioning to dilemma and then staghunt. As $\Delta\phi$ increases from $\pi$ to $2\pi$, this process reverses.

These different game types have multiple roles in the subsequent analysis. First, they provide interpretability of the dynamics in different regions of parameter space. The three-dimensional parameter space depicted in Fig 2 hosts a wide variety of dynamics with varying invasion possibilities, fixation times, and stabilities. However, by leveraging existing knowledge of classic

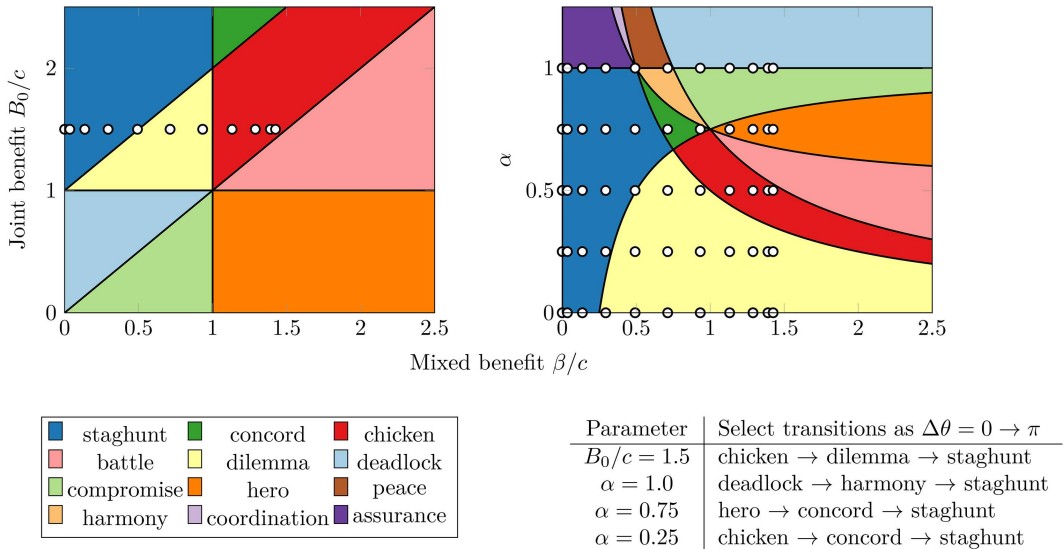

**Fig 2. Payoff asymmetry enriches neural interactions well beyond the classic prisoner's dilemma game type. Region plots illustrate the diverse range of game types that neural populations can engage in during evolutionary dynamics.** Slices of the three-parameter mixed *CN* game-type phase diagram in the **(A)** $\beta - B_0$ plane ($\alpha = 0.5$) and **(B)** $\beta - \alpha$ plane ($B_0/c = 1.5$). For two players with phase difference $\Delta\phi$, the mixed benefit is $\beta = \beta_0[1 + \cos(\Delta\phi)]/2$. The legend displays the game type corresponding to each color. The white dots represent the $m = 20$ potential phase differences as well as the restriction **(A)** $B_0/c = 1.5$ or **(B)** $\alpha \in [0, 0.25, 0.5, 0.75, 1]$. **(C)** A table showing key transitions between game types to clarify the reading of (A) and **(B)**. The left column describes the fixed parameter, with $B_0/c$ corresponding to white dots on (A) and $\alpha$ to white dots on **(B)**. As the phase difference $\Delta\phi$ increase from 0 to $\pi$, $\beta = \beta_0 f(\Delta\phi)$ decreases and the dots move to the left, tracing the games in the right column.

two-player game types, we can understand the features of each region. Second, they allow us to tailor the model for examining different interaction classes. Prior studies on neurons frequently use prisoner's dilemma type games; our model allows for targeting the "dilemma" (yellow) region by tuning the parameters $B_0/c$, $\beta_0/c$, and $\alpha$ (*e.g.,* the bottom two lines of white dots in Fig 2B with $\alpha = 0$ and $\alpha = 0.25$) Alternatively, it might be desirable to analyze the behaviour of game types beyond the prisoner's dilemma, which may exhibit exponentially long fixation times [29] such as chicken, hero, harmony, and battle (the warm-colored game types). We see that is possible for our values of $B_0$ and $\beta_0$ when $\alpha = 0.5$ or $\alpha = 0.75$ (white dots intersect the orange "hero" and red "chicken" regions in Fig 2B). Therefore, while there is no intrinsic, biological representation of "game types" at the biochemical level, they offer a valuable interpretability tool and a useful way to target model outcomes.

## 2.3 Simulation parameters

We have now introduced a number of parameters: $B_0$, $\beta_0$, $c$, $\alpha$, $m$. In addition, our model uses the usual evolutionary game theory parameters: selection strength $\delta$ and mutation rate $\mu$ (*cf.,* Section 4.3). Therefore, it is helpful to quickly review the dependencies between the parameters and their importance. As discussed in Section 4.3, the evolution is governed by a Moran process with exponential fitness $f_i$:

$$\delta c \begin{bmatrix} \frac{B_0}{c}f(0) - 1 & 2\alpha\frac{\beta_0}{B_0}\frac{B_0}{c}f(\Delta\phi) - 1 \\ 2(1-\alpha)\frac{\beta_0}{B_0}\frac{B_0}{c}f(\Delta\phi) & 0 \end{bmatrix}$$

Thus, we can see that the four combinations $B_0/c$, $\beta_0/B_0$, $\alpha$, and $\delta c$ serve to define the relative strengths of four combinations *CC*, *CN*, *NC*, and *NN*. Therefore, the dynamics are wholly controlled by the parameter combinations $B_0/c$, $\beta_0/B_0$, $\alpha$, and $\delta c$, in addition to the aforementioned mutation rate $\mu$ and discretization number $m$.

The ratio $B_0/c \approx 2.1$ is the critical joint-benefit benefit strength at which the communicative fraction equals 1/2 when in the absence of asymmetry(*i.e.*, $\alpha$ = 1/2) [5]. Therefore, we follow [5] and choose $B_0/c$ = 1.5 slightly less than the critical point to slightly bias the system towards non-communicativeness. Likewise, we choose $\beta_0/B_0$ = 0.95 since we biologically expect mixed *NC* communication to be less rewarding than joint *CC* communication. Both the mutation rate $\mu$ and selection strength $\delta$ are chosen to be small to allow for analytic approximations (see section 2 of S1 Text in S1 Appenidx). While we fix a common set of parameters in our results for simplicity, we also performed a sensitivity analysis in Section 4.9 to analyze the influence of varying these parameters. The main results (the chimera-like index $\chi$) are largely insensitive to changes in most of these parameters. The only sensitive parameter is the selection strength $\delta$, with stronger $\delta$ increasing the chimera-like effect. However, we retain the smaller $\delta$ for analytic tractability. The $m = 20$ discrete phases approximate a continuous phase distribution; Fig 10 depicts a convergence analysis showing that the communicative fraction has converged already for $m = 20$, verifying that this is an acceptable approximation.

Unless otherwise specified, each of the following simulations use $m = 20$ phases, selection strength of $\delta$ = 0.2, mutation rate of $\mu = 1 \times 10^{-4}$, cost $c$ of 0.1, maximum joint benefit $B_0 = 0.15$, and maximum mixed benefit $\beta_0$ of 0.95 $B_0$.

Most runs use $8 \times 10^6$ time steps to ensure that we have sampled sufficiently many mutation periods $1/\mu$ (though time series plots show a temporal subset for visual clarity). With $\mu = 1 \times 10^{-4}$, this means we sample approximately 800 mutations; this high number of mutations allow us to efficiently sample the available parameter space. The plots of communicative fraction $f_{comm}$ use a longer runtime of $2 \times 10^8$ to collect additional statistics, drive down the sample variance, and provide tighter error-bounds on the comparison to analytic theory.

## 2.4 Complete graphs

First, we will explore the influence of the newly introduced asymmetry on a $N = 20$-player, well-mixed population. Fig 3A compares the frequency of communicative strategies $f_{comm}$ to the maximum joint benefit $B_0$. The marks represent the simulation results and the lines represent the full analytic result (Eq. 10 in S1 Text in S1 Appenidx) for a well-mixed population. The $B_0$ step size is 0.04, and the simulations ran for $2 \times 10^8$ time steps. The results are averaged over 10 seeds and the vertical error bars show the standard deviations. In general, $f_{comm}$ is low for small $B_0$, rises to $f_{comm} = 0.5$ at some break-even $B_0$, and plateaus to $f_{comm} \approx 1$ for large $B_0$. We can validate our model by comparing our $\alpha$ = 0.5 case with a previous study [5] to observe qualitatively similar results, with our break-even $B_0 \approx 0.21$ corresponding to their $B_0 = 2(N-1)c/(N-2) \approx 0.21$ break-even condition. We also see that increasing (decreasing) the asymmetry $\alpha$ dilates (stretches) this sigmoid function in the $B_0$ direction. This $\alpha$-dependence is reasonable, as increasing $\alpha$ corresponds to biasing the payoff in a mixed *CN* interaction towards the communicative partner.

While Fig 3A displays the time-averaged system state, it is also useful to investigate the time-dependent variations. Figs 3B to 3D depict the frequency of communicative strategies $f_{comm}$ as scatter plots of time on the left vertical axis for different values of the asymmetry $\alpha$. These time-series points are color-coded grey if all players are communicative or black if all players are non-communicative; otherwise, the points are colored according to the plurality mixed game type, as indicated in the legend. On the right vertical axes, magenta line plots depict the order parameter $\rho \in [0, 1]$ given by Eq. 4. The order parameter and communicative frequency are shown for a single seed to demonstrate the mutation-induced jumps for a single simulation. The mutation rate $\mu = 1 \times 10^{-4}$ means that a mutation occurs, on average, every $1 \times 10^4$ time steps; for the $8 \times 10^5$ time steps, we therefore expect 80 mutations. These mutations cause the observed dips in the order parameter. But we note that there are significantly fewer than 80 dips shown. The quick extinction of mutant invaders means that they are often missed when we 80×-decimate the time-series for plotting, only observing $\approx 6$ visible mutations.

Fig 3B shows the time-variation when the system heavily favors non-communicative players with $\alpha$ = 0. After an initial disordered, dilemma-type game (yellow), the system synchronizes in the non-communicative regime (black line at $f_{comm} = 0$) with only brief, dilemma-type game excursions at $t \approx 2.5 \times 10^5, 4.5 \times 10^5$. Fig 3C depicts an asymmetry $\alpha$ = 0.75

PLOS Computational Biology

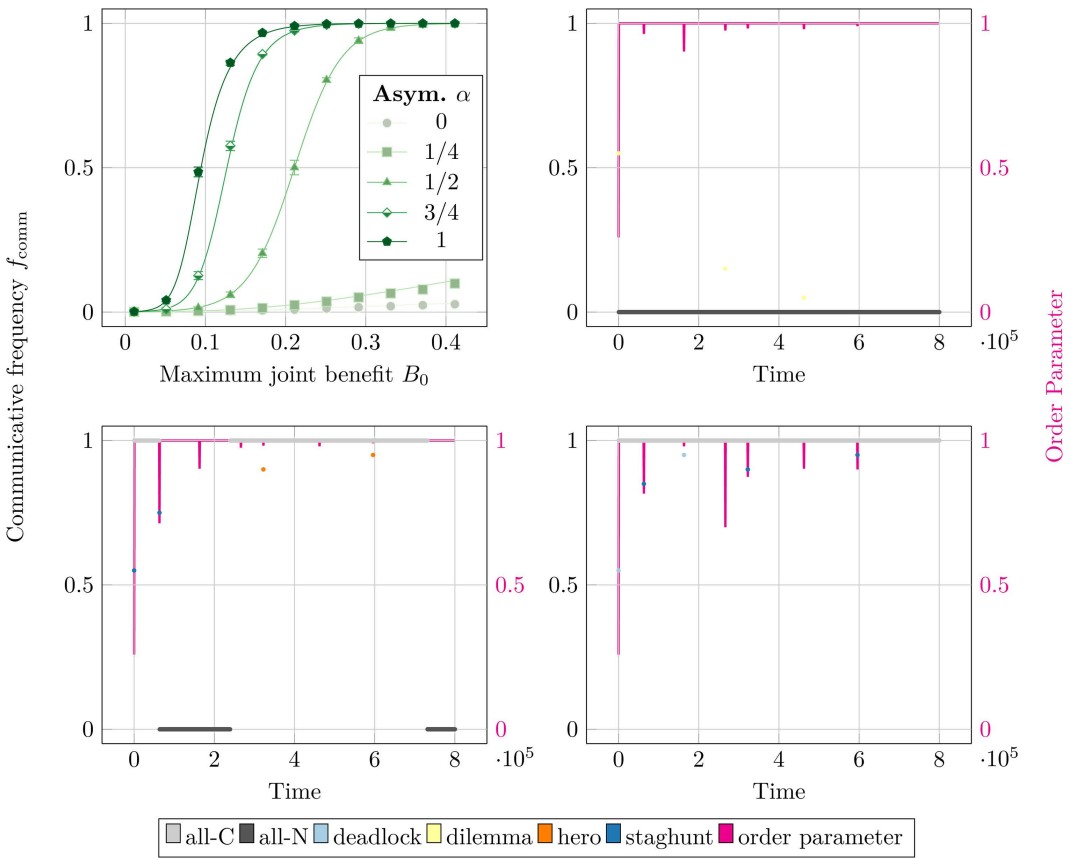

**Fig 3. Impact of symmetry breaking on neural synchronization in well-mixed populations.** Communication frequency $f_{comm}$ for the well-mixed topology. **(A)** Time-averaged $f_{comm}$ as a function of the maximum joint benefit $B_0$ for different values of the payoff asymmetry $\alpha$. The lines represent the theory predictions, the marks represent the simulation results averaged across 10 seeds, and the vertical error bars denote the standard-deviation across the 10 runs. **(B)**–**(D)** Scatter plots where the left vertical axes show the instantaneous $f_{comm}$ as a function of time and are color-coded according to the plurality mixed game type as indicated in the legend. The right vertical axes give the order parameter $\rho$ (Eq. 4), in magenta, as a function of time. The asymmetry is **(B)** $\alpha = 0$, **(C)** $\alpha = 0.75$, and **(D)** $\alpha = 1$.

that moderately encourages communicativeness. The system is synchronized in a communicative state (gray line at $f_{comm} = 1$) 69% of the time, with unstable excursions to synchronized, non-communicative states (black line at $f_{comm} = 0$) and disordered, hero (orange) and staghunt (dark blue) game types. This 69% communicative rate is similar to the expected 75% from Fig 3A (for $B_0 = 0.15$ and $\alpha = 0.75$), with small differences likely due to the short timespan shown in Fig 3C. Finally, Fig 3D shows the $\alpha = 1$ case where communication is heavily incentivized. Like the (B) $\alpha = 0$ case, the system is virtually always synchronized in the communicative regime, with only brief excursions to deadlock (light blue) or staghunt (dark blue) games types. We note that all three (B–D) scenarios are virtually always synchronized; the order parameter $\rho$ (magenta) is almost always $\rho = 1$, with only occasional dips due to mutations.

## 2.5 *C. elegans* graphs

Here, we consider the weighted, directed hermaphroditic *Caenorhabditis elegans* (*C. elegans*) chemical connectome [21]. For reference, Fig 4D shows the $N = 300$ nodes and the directed edges (we exclude the two unconnected neurons CANL/CANR). We treat all *Caenorhabditis elegans* (*C. elegans*) neurons identically and do not differentiate between sensory neurons, interneurons, and motor neurons. The nodes are colored according to their strategy at a particular time step,

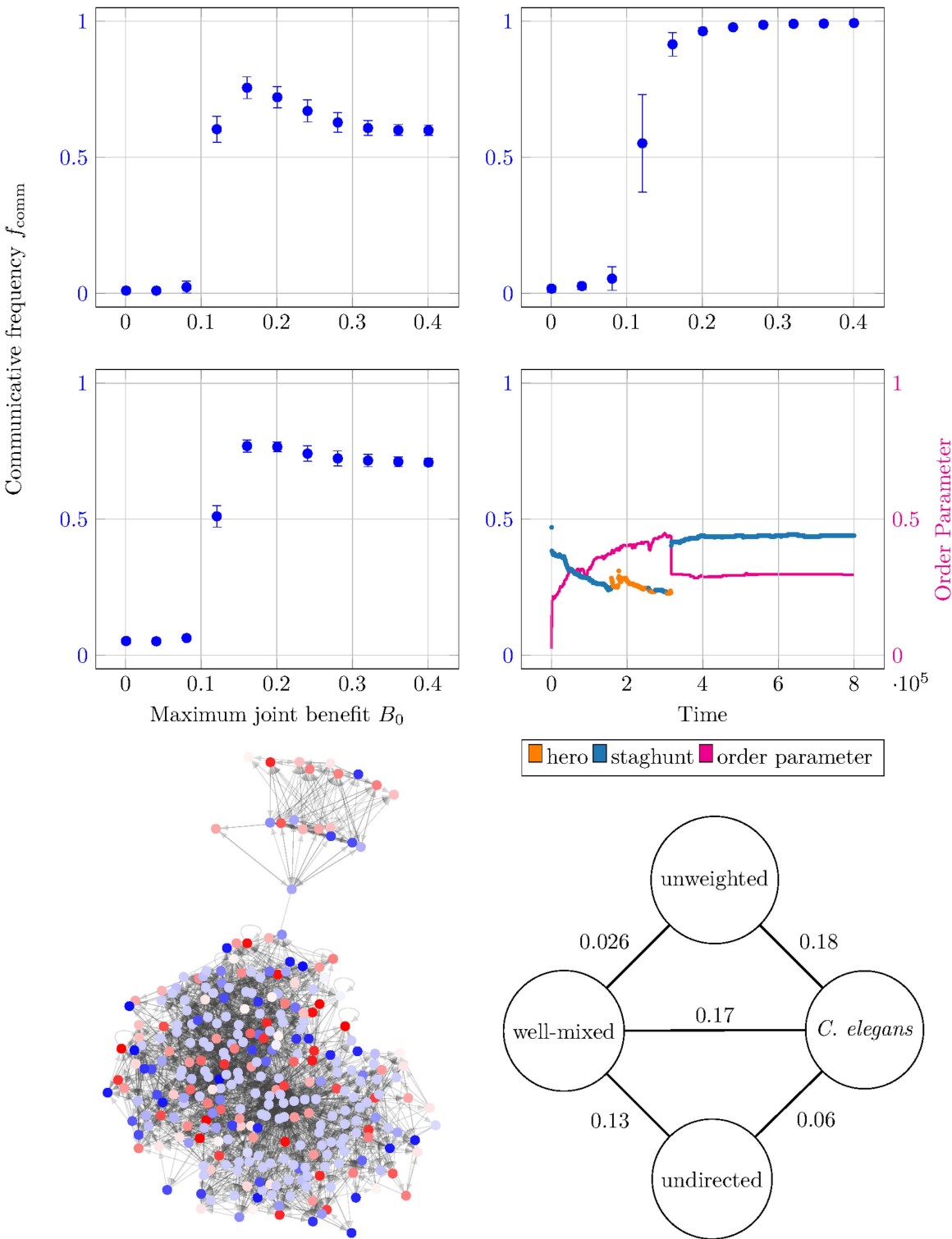

**Fig 4. Rise of chimera states in *Caenorhabditis elegans* (*C. elegans*) neural network. (A)–(C)** Time-averaged fraction of players that are commu-nicative as a function of the maximum joint benefit $B_0$. The network topologies are the **(A)** weighted, directed *Caenorhabditis elegans* (*C. elegans*) con-nectome, **(B)** unweighted, directed *Caenorhabditis elegans* (*C. elegans*) connectome, and **(C)** weighted, undirected *Caenorhabditis elegans* (*C. elegans*) connectome. The results are averaged over 10 seeds and the error bars depict the standard deviations. **(D)** The network topology for the *Caenorhabditis*

elegans (*C. elegans*) $N = 300$ weighted, directed connectome with asymmetry $\alpha = 0.75$. The colors represent the $2m = 40$ strategies at a particular time step; blue colors are communicative, red colors are non-communicative, and shades represent different phases $\phi$. **(E)** Scatter plot with the same axes and coloring as Figs 3B to 3D showing the communicative frequency, plurality mixed game-types, and order parameter. **(F)** A flowchart showing the pairwise standard deviation between the various communicative fractions $f_{\text{comm}}$: "well-mixed" is Fig 3C, "unweighted" is **(B)**, "unundirected" is **(C)**, and "*C. elegans*" is **(A)**.

with blues representing communicative strategies and reds representing non-communicative ones, and different shades corresponding to different phases $\phi$. We note the chimera-like character of the large, synchronized group of red nodes coexisting with disordered neighboring nodes.

Next, to quantify the observations of Fig 4D, Fig 4A shows the fraction of players using communicative strategies $f_{\text{comm}}$ averaged across the entire simulation of $2 \times 10^8$ time steps as a function of the maximum joint benefit $B_0$ in 0.04 steps for the *Caenorhabditis elegans* (*C. elegans*) connectome network topologies. All subplots use an asymmetry of $\alpha = 0.75$. Additionally, the $B_0$-dependent plots (A–C) show the average across 10 seeds; error bars represent the standard deviation.

We note that the behaviour of the (A) *Caenorhabditis elegans* (*C. elegans*) case is qualitatively distinct from the $\alpha = 0.75$ well-mixed case in Fig 3A. The communicative fraction $f_{\text{comm}}$ is flatter for $B_0 \leq 0.08$, has a steep jump to 0.77 at $B_0 = 1.6$, and decreases to a horizontal asymptote around 0.60. We can isolate the cause of this deviation from the Fig 3A well-mixed behavior by looking at variations to the Fig 4A *Caenorhabditis elegans* (*C. elegans*) network topology. First, the (B) *directed*, unweighted connectome is qualitatively similar to the (A) well-mixed case with a monotonic increase from low communicativeness for $B_0 < 0.1$ to full communicativeness for $B_0 \geq 0.2$. This implies that directedness plays only a small role in the qualitative shape of the (A) full *Caenorhabditis elegans* (*C. elegans*) case. Conversely, the (C) *weighted*, undirected connectome looks similar to the (A) full *Caenorhabditis elegans* (*C. elegans*) case, displaying the same plateau at $f_{\text{comm}} \approx 0.7$, though the peak around $B_0 = 0.15$ is less pronounced. This similarity implies that the connectome's edge weights cause most of the deviation between the (A) full *Caenorhabditis elegans* (*C. elegans*) case and the Fig 3A well-mixed case.

We can further quantify the relative influence of weightedness and directedness by comparing (A–C) with the Fig 3C well-mixed case. Fig 4F shows the pairwise standard deviations calculated between the well-mixed, unweighted, undirected, and full *Caenorhabditis elegans* (*C. elegans*) cases. In particular, we see that the unweighted *vs.* *Caenorhabditis elegans* (*C. elegans*) standard deviation is three times larger than the undirected *vs. Caenorhabditis elegans* (*C. elegans*) standard deviation. Therefore, this implies that the weighted (undirected) edges have three times the influence as the directed (unweighted) edges in the full *Caenorhabditis elegans* (*C. elegans*) communicative frequency profile.

We can also investigate the time-evolution of the *Caenorhabditis elegans* (*C. elegans*) system using the same parameters as the Figs 3B to 3D well-mixed case but with the *Caenorhabditis elegans* (*C. elegans*) connectome graph. Compared to the well-mixed case, the (E) *Caenorhabditis elegans* (*C. elegans*) case depicts a far more heterogeneous population. Here, the population never stabilizes to a fully communicative or non-communicative state. Instead, its communicative frequency stays between 22% to 47%, and its mean $f_{\text{comm}} \approx 37\%$ is smaller than the expected $f_{\text{comm}} \approx 70\%$ from Fig 4A with $B_0 = 0.15$; the discrepancy likely comes from the stochasticity in this small, $8 \times 10^5$ time-step subset. Similarity, the order parameter's smaller value of 2% to 45% indicates far less synchronization than the well-mixed case. And while the Fig 3C well-mixed setup displays hero, staghunt, and concord plurality mixed game-types, the Fig 4E *Caenorhabditis elegans* (*C. elegans*) setup only displays hero and staghunt games. Finally, the *Caenorhabditis elegans* (*C. elegans*) case is less stable to mutations than the Fig 3C well-mixed case: instead of stable synchronized states with transient impulses, the *Caenorhabditis elegans* (*C. elegans*) case depicts disordered states with discontinuous offsets.

## 2.6 Chimera-like index

Time-lapse animations of the system's time evolution show a subset of the nodes exhibiting high synchronicity with others remaining disordered: this is characteristic of a chimera state. Animations depicting this phenomena are available as supporting information (S1 Video, S2 Video, and S3 Video), and an interactive website for exploring the full data set through plots and animations is provided in the "Code availability". In order to quantify this chimera-like effect, we will investigate the chimera-like index $\chi$ (Eq. 6) and metastability index $\lambda$ (Eq. 7). As discussed in Section 4.8, the chimera-like index measures the coherence difference between communities of players, is equal to the time-averaged community variance, and has a theoretical maximum value of $M/[4(M-1)] = 0.3$ [17] for our $M = 6$ communities. Conversely, the metastability index represents how often the system transitions between synchronicity and disorder, is equal to the community-average of the temporal variance; it is 0 for communities that are always synchronized or always disordered, and increases to 0.25 for communities spending equal time fully synchronized and fully disordered. Fig 5 shows the (A) chimera-like index $\chi$ and (B) metastability index $\lambda$ as functions of the asymmetry $\alpha$. These simulations ran with

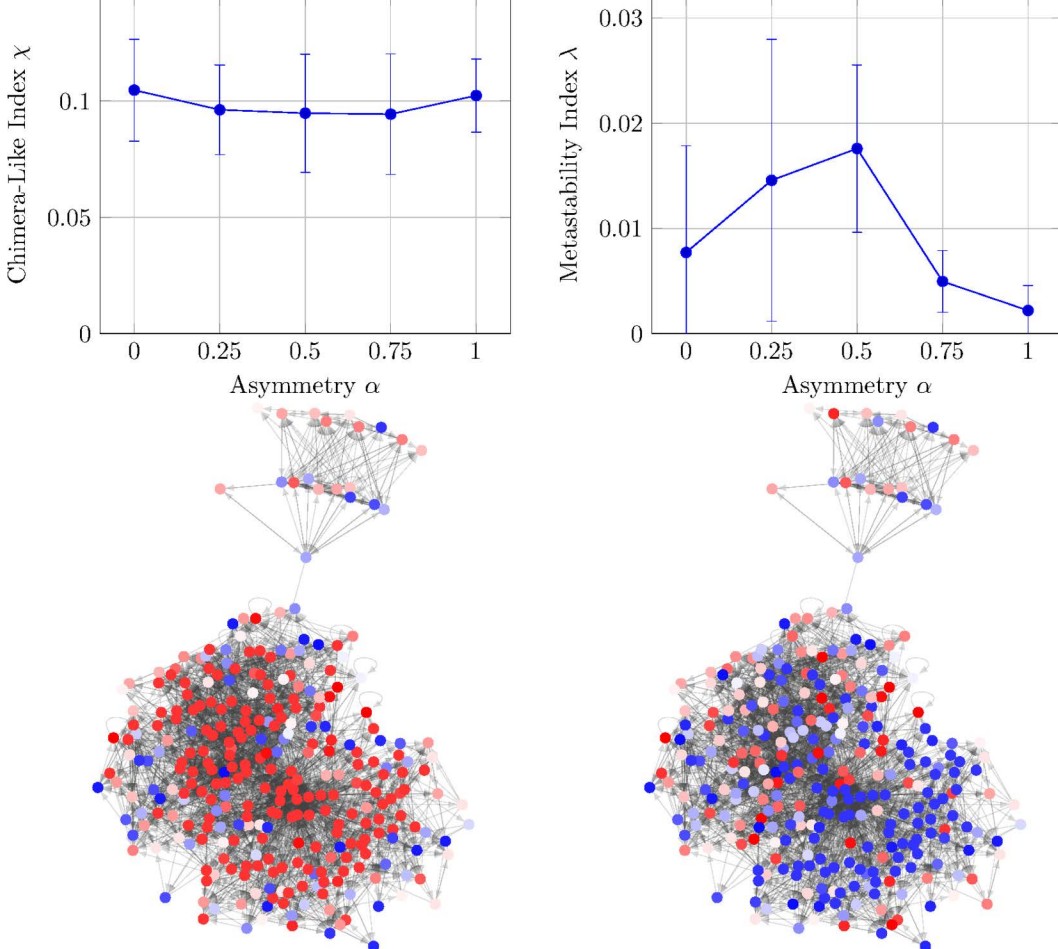

**Fig 5. Characterizing chimera states. (A)** The chimera-like index (Eq. 6) and **(B)** metastability index (Eq. 7) as functions of the asymmetry $\alpha$ for the weighted, directed *Caenorhabditis elegans* (*C. elegans*) connectome. To calculate the indices, we split the graph into two communities using the constant Potts model (*cf.*, Section 4.8). Results were averaged over 10 seeds, and error bars depict the standard deviation. **(C)–(D)** A snapshot of the *Caenorhabditis elegans* (*C. elegans*) connectome where blue colors are communicative, red colors are non-communicative, and shades represent different phases $\phi$. The snapshots show a particular seed and time step with asymmetry **(C)** $\alpha$ = 0 or **(D)** $\alpha$ = 1.

the same parameters as the Figs 4A to 4C *Caenorhabditis elegans* (*C. elegans*) time-series data averaged over 10 seeds with the error bars depicting the standard deviations.

The metastability displayed in Fig 5 is less than 0.03, much smaller than the maximum of 0.25. This implies that the system has low metastability and spends most of its time at a nearly constant synchronicity $\rho_m(t)$. Furthermore, while the metastability is highest (more stability variations) t $\lambda \approx 0.02$ for the $\alpha = 0.25, 0.5$ cases, the metastability drops for larger $\alpha$, always staying below 0.01. Conversely, the chimera-like index $\chi$ indicates a high chimeric quality with values ($\chi \approx 0.1$) a third of the theoretical maximum (0.3). Given that we average the chimera-like index over time, the system's deviation from a complete chimera state arises from both imperfect separation of the coherent/disordered populations as well as time fluctuations in the chimeric quality.

The chimera-like index depicted in Fig 5 is a time-averaged statistical quantity, but the interplay between synchronicity and disorder can also be understood as a function of time. In particular, the chimera-like states are characterized by some communities with low order parameter $\rho_m$ (Eq. 5) and some with high order parameter. Therefore, Fig 6 shows the community parameter for each of the six communities as a function of time. These are the same six parameters as those used in Fig 5 which were calculated using a constant Potts model.

Fig 5 clearly depicts the order-parameter variability between the six communities. Communities 1–4 are mostly disordered, with order parameter ≤0.6, while communities 5 and 6 are very synchronized, with order parameters ≥0.95. This bifurcation into ordered and disordered subgroups is a key characteristic of chimera-like states, showing that the chimera-like index in Fig 5 is depicting a real separation of community coherence.

## 2.7 Game types

Next, we seek to understand the types of games that nodes play during the population's evolution. In Fig 7, we investigate the plurality game type amongst all player interactions at each time step (*cf.,* Section 4.6). Fig 7 shows the fraction of time that a given game type is the plurality for different values of the asymmetry $\alpha$ for (A) $N = 20$ well-mixed, and (B) *Caenorhabditis elegans* (*C. elegans*) connectome network topologies. The fractions are averaged over 10 seeds. The (A) well-mixed case fully-synchronized to all-communicative or all-noncommunicative for over 99.6% of the runtime across every asymmetry $\alpha$, as corroborated by the order parameter $\rho \approx 1$ in Figs 3B to 3D. Furthermore, the $\alpha$-dependence of the

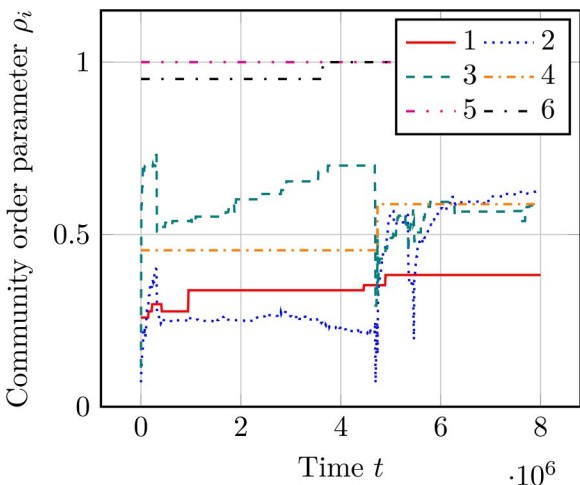

**Fig 6. Intra-community variation.** The community-dependent order parameter $\rho_m(t)$ (Eq. 5) as a function of time. These six communities are the same as those use in Fig 5 calculated using the constant Potts model (*cf.,* Section 4.8). The asymmetry is $\alpha = 0.75$, and the remaining parameters are identical to those used in Fig 4.

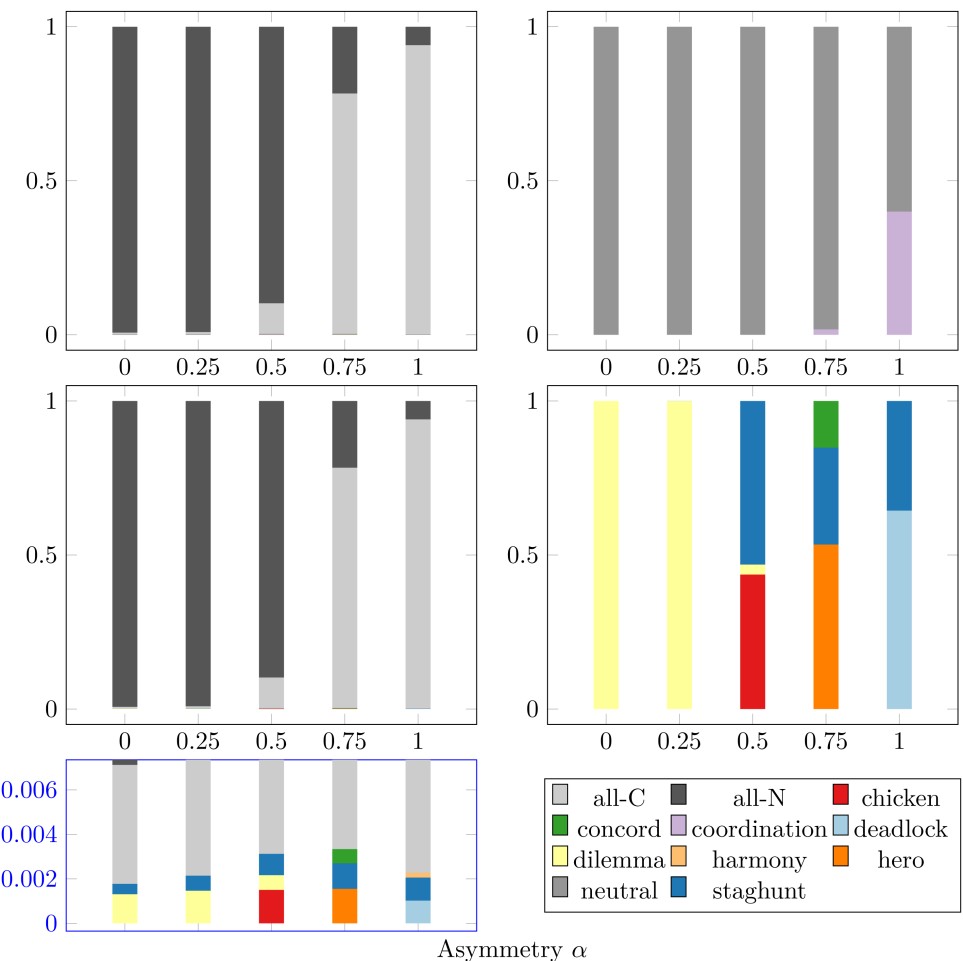

**Fig 7. Influence of payoff asymmetry on game types played.** The plurality game type amongst all player interactions expressed as a fraction of all games played for different values of the asymmetry $\alpha$. The game types are color-coded according to the legend; additionally, "all-C" and "all-N" represent when the population is entirely synchronized to communicativeness or non-communicativeness, respectively. The network topologies are the **(A)** $N = 20$ well-mixed population and **(B)** weighted, directed *Caenorhabditis elegans* (*C. elegans*) connectome. Similarly, **(C)**, **(D)** depict the plurality game type when only considering mixed game types for the **(C)** $N = 20$ well-mixed population and **(D)** weighted, directed *Caenorhabditis elegans* (*C. elegans*) connectome. A blue inset below the **(C)** well-mixed panel shows the magnified lower region. All fractions were averaged across 10 seeds.

communicative synchronization ("all-C") fraction approximates the communicative frequency $f_{comm}$ in Fig 3A for $B_0 = 0.15$. In contrast, the (B) *Caenorhabditis elegans* (*C. elegans*) system is never synchronized, but instead is virtually always dominated by coordination-type games (when both players are communicative, *CC*) and neutral-type games (when both players are non-communicative, *NN*).

In order to investigate the other game types involved, we can also consider the plurality *mixed* game type (*cf.,* Section 4.6). The (C) well-mixed case only changes for the < 1% of the time when unsynchronized. In order to better observe these games types, a blue inset below Fig 7C magnifies these small fractions. In contrast, the (D) *Caenorhabditis elegans* (*C. elegans*) system shows a variety of most-frequent game types that vary based on the asymmetry: dilemma for $\alpha = 0, 0.25$, staghunt for $\alpha = 0.5$, hero for $\alpha = 0.75$, and deadlock for $\alpha = 1$. Additionally, the *Caenorhabditis elegans* (*C. elegans*) system displays some game-type heterogeneity, with a second game type (staghunt) being the plurality 1% to 18% of the time for asymmetries $\alpha \geq 0.5$. Finally, we note that the *Caenorhabditis elegans* (*C. elegans*) mixed game distributions are

approximately the same as the *disordered* (*i.e.,* states other than "all-C" or "all-N") plurality mixed games in the Fig 7C well-mixed setup (*cf.,* blue inset) for each $\alpha$. Overall, we see that graph structure decreases synchronization and asymmetry influences the dominant game types, which, in turn, underpin the formation of chimera states.

In addition to the time-averaged plurality game type depicted in Fig 7, we can also look at the fine-grained distribution of all games played at a single time step. Fig 8 shows the types of games played as a fraction of all games among all 3,707 directed edges of the *Caenorhabditis elegans* (*C. elegans*) connectome. The snapshot was taken at time $t = 4 \times 10^6$ of the simulation. (A) depicts the fraction of each game, including both mixed games (*CN* interactions) as well as *CC* interactions (coordination-type games) and *NN* interactions (neutral-type games). (B) shows the same result restricted to *CN*/*NC* mixed games.

The set of all games in Fig 8 shows some interesting differences when compared to the plurality games in Fig 7. First, the full set of games in Fig 8A displays a far greater variety of games than the neutral and coordination games depicted in the plurality metric Fig 7B. This shows that these mixed games—dilemma, staghunt, chicken, hero, deadlock, and concord—are present in the population, even if they are never frequent enough to become the plurality. Similarly, when just looking at mixed-type games in Fig 8B, we see staghunt games for $\alpha \leq 0.25$ and harmony games for $\alpha = 1.0$ that never became a plurality in Fig 7D. For $\alpha = 0.5, 0.75$, we see much greater similarity between Fig 8B and Fig 7D, implying each game type present becomes the plurality, though not necessarily with temporal frequency equal to its spatial frequency. Thus, we see that the rich array of game types accessible to the system in the extended $B_0 - \beta_0 - \alpha$ phase space of Fig 2 is thoroughly explored, even when all of the game types are not necessarily visible in the plurality metric of Fig 7.

## 3 Discussion

We can validate our Fig 7C well-mixed time-series results by comparing the $\alpha = 0.5$ case to previous studies [5]. Their weak-selection $\delta = 0.2$ results also show the system spending virtually all of its time in a synchronized state. When not synchronized, their system had plurality chicken (denoted "snowdrift" therein) and staghunt (denoted "coordination", not to be confused with the coordination-type game here) games. Indeed, Fig 2A shows that staghunt, dilemma, and chicken would be the only game types accessible for $\alpha = 0.5$, which is corroborated in our simulations by the inset below Fig 7C.

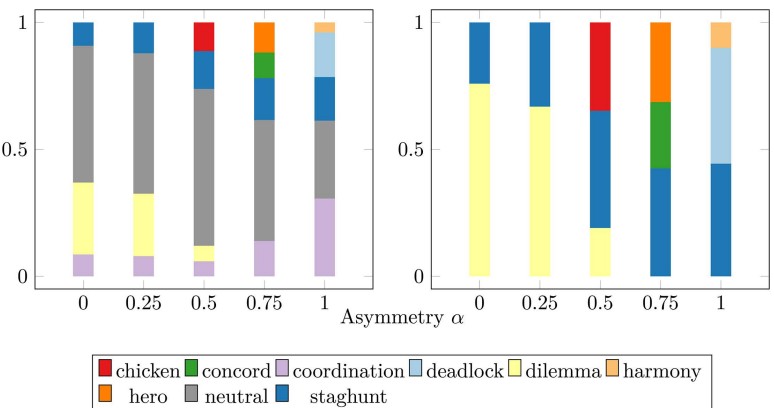

**Fig 8. Game type diversity amongst *Caenorhabditis elegans* (*C. elegans*) neurons.** The game types at a single time step amongst all player interactions expressed as a fraction of all games played for different values of the asymmetry $\alpha$. The game types are color-coded according to the legend; additionally, "all-C" and "all-N" represent when the population is entirely synchronized to communicativeness or non-communicativeness, respectively. The network topologies is the weighted, directed *Caenorhabditis elegans* (*C. elegans*) connectome. **(A)** depicts the fraction played of each game type, including "communicative" games between *CC* pairings and "neutral" games between *NN* pairings, while **(B)** depict the game types when only considering mixed game types of *CN* pairings. The snapshot was taken at time $t = 4000000$. All fractions were averaged across 10 seeds.

Next, we will discuss the observed chimera states. Another study of the *Caenorhabditis elegans* (*C. elegans*) connectome also found chimera-like states using an entirely non-game-theory tool, modular neural networks [24]. This study applied a Hindmarsh-Rose model of ODEs to the *Caenorhabditis elegans* (*C. elegans*) connectome with both chemical and electrical connections. The authors split the connectome into six communities using a walktrap algorithm and used strong, linear electrical connections within communities and weaker, nonlinear chemical connections between communities. In contrast, our model uses nonlinear chemical connections between all neurons of the experimentally measured chemical connectome [21], but does not include the (distinct) electrical connectome. The authors found a relatively large metastability index $\lambda$, up to $\lambda = 0.07$ for strong inter-community electrical connections. In contrast, our model generated a smaller metastability index $\lambda \leq 0.03$; this is likely due to our homogenous synapse type between- and within-communities, suppressing the heterogeneity necessary for metastability. Furthermore, the authors also observed chimera-like states with a maximum chimera-like index of 0.12. The agreement with our chimera-like index of $\chi \approx 0.08 \text{ to } 0.12$ for $\alpha = 0.5$, is encouraging; the agreement in $\chi$ between two very different models of neurons suggests that chimera states these chimera states are a robust feature of the *Caenorhabditis elegans* (*C. elegans*) connectome independent of the model.

Furthermore, our model offers two novel improvements over a Hindmarsh-Rose model: interaction-interpretability and computational ease. The interpretation of neuron interactions using classical 2-player game types offers insight into the different behaviors displayed in different regions of the parameter space. Additionally, compared to a system of ODEs like the Hindmarsh-Rose model, our discrete-strategy and -phase EGT model is less computationally demanding. This computational advantage enables wide parameter sweeps (see, *e.g.,* Section 4.9) for determining which parameter combinations produce the strongest chimera-like signals. Given the hypothesized role of chimera-like states in cognition [19,20], this computational tool could enable correlating experimental statistics of *Caenorhabditis elegans* (*C. elegans*) chimera states with single-neurons payoffs $B_0$, $\beta_0$, *etc.*, improving neuron characterization and enable further *in silico* studies.

Chimera states of networked oscillators exhibit coexisting synchronized and disordered populations, are present in brains [18,19], and may be pivotal in human cognition [18]. Prior studies [11] investigated chimera-like brain states in *Drosophila melanogaster* (*D. melanogaster*) using sinusoidally-coupled Kuramoto oscillators, a frequent model for neuron dynamics [10]. However, the small-scale evolutionary factors leading to chimera-like states remains poorly understood. In this paper, we extended the evolutionary Kuramoto model to include weighted, structured interaction graphs and an asymmetry between the communicator and non-communicator payoffs. The asymmetry represents differences in communicative and non-communicative biochemical processes, and its role in a well-mixed population showed that that increasing (decreasing) the asymmetry inhibits (promotes) communicativity. Additionally, our extension of an evolutionary Kuramoto model to weighted, directed graphs enabled analysis of realistic brain networks like the *Caenorhabditis elegans* (*C. elegans*) chemical connectome. Simulations of *Caenorhabditis elegans* (*C. elegans*) neurons revealed that the graph's weightedness has a much stronger influence on the communication frequency than the graph's directedness does. Finally, while the well-mixed players have homogeneous populations that occasionally switch between synchronized communicative and non-communicative states, the *Caenorhabditis elegans* (*C. elegans*) population remains far more heterogenous with a stable, chimeric mix of disordered game types.

One potential limitation of this work is the identical treatment of different neuron types. Potential extensions could modulate the $B_0/c$, $\beta_0/c$, and $\alpha$ parameters of the different neuron types separately to better account for the specialization of these cells. Additionally, future work will focus on applying the EK model to families of generative brain network models [30]. The limited number of empirical graphs analyzed in this study hinders the identification of exact relationships between the communicative fraction and graph properties such as edge degree, weight, and directionality. However, applying the model to parameterized families of graphs could allow for fine-tuning the parameters to extract these relations. Additionally, since this study only considers a single species' connectome, it is unclear if our findings regarding the dependence of chimeric activity on graph structure are generalizable. Therefore, it is of interest to investigate this connection further by applying our EK setup to other model organisms, such as that of *Drosophila melanogaster* [31]. Our

result on the importance of edge-weights on suppressing communication will preferentially enhance chimera-like states in connectomes with larger edge-weight heterogeneity. On the contrary, the relatively small effect of edge directivity implies that connectomes with a large number of bi-directional pairings is likely to have weaker chimera-like states. Finally, the observation that stronger selection strength $\delta$ enhances the chimera-like index suggests that connectomes with a greater proportion of high-degree nodes with have stronger chimera-like states: in our model, stronger selection $\delta$ biased higher in-degree nodes (since we used total payoff instead of average payoff); thus, connectomes with naturally more high-degree nodes would be comparable to our strong-selection $\delta$ case with robust chimera-like states.

A key takeaway from this study was the importance of edge weight on communicativity; this has important parallels to neural computing, where edge weights are a primary driver of functionality. Additionally, the observation of chimeric states arising from such simple neurogame-theoretic models implies that the nature of neuron interactions is likely a key component in producing these critical brain states. We demonstrated that introducing weighted, directed connectivity in the *Caenorhabditis elegans* (*C. elegans*) connectome graph was sufficient to generate robust chimera states modulated by payoff asymmetry between communicators and non-communicators. Overall, evolutionary graph theory allowed us to connect low-level payoff details for individual neurons to high-level phenomena such as chimera-states, and this model could serve as a valuable computational framework for clarifying the influence of network structure on neural dynamics.

## 4 Methods

### 4.1 Game setup

We model the system of evolving, coupled oscillators by discretizing the $2\pi$ phase angle into $m$ discrete phases $2\pi j/m$ for $j \in 0, \ldots, m-1$. The game's strategy space is the outer product of the $m$ phases ($m = 20$ for our simulations) and 2 communicative choices, communicative $C$ and non-communicative $N$. For a given pair of phases, $\phi_i$ and $\phi_j$, the game is specified by the payoff matrix. If one player is communicative and the other is non-communicative ("mixed game", $CN$ or $NC$), the payoff matrix is

$$
\begin{array}{c}
 \\
C, \phi_i \\
N, \phi_j
\end{array}
\begin{array}{c}
\quad C, \phi_i \qquad\qquad N, \phi_j \\
\left[\begin{array}{c|c}
B_0 f(0) - c & 2\alpha\beta_0 f(\Delta\phi) - c \\
\hline
2(1-\alpha)\beta_0 f(\Delta\phi) & 0
\end{array}\right],
\end{array}
\tag{1}
$$

if both players are communicative ($CC$), the matrix is

$$
\begin{array}{c}
 \\
C, \phi_i \\
C, \phi_j
\end{array}
\begin{array}{c}
\quad C, \phi_i \qquad\qquad C, \phi_j \\
\left[\begin{array}{c|c}
B_0 f(0) - c & B_0 f(\Delta\phi) - c \\
\hline
B_0 f(\Delta\phi) - c & B_0 f(0) - c
\end{array}\right],
\end{array}
\tag{2}
$$

and if both are non-communicative ($NN$), the matrix is

$$
\begin{array}{c}
 \\
N, \phi_i \\
N, \phi_j
\end{array}
\begin{array}{c}
\quad N, \phi_i \quad N, \phi_j \\
\left[\begin{array}{c|c}
0 & 0 \\
\hline
0 & 0
\end{array}\right],
\end{array}
\tag{3}
$$

with $f(\Delta\phi) = [1 + \cos(\phi_j - \phi_i)]/2$ (Fig 1C). Here, $B_0$, $\beta_0$, $\alpha$, and $c$ are non-dimensional, fixed parameters defining the game. The cost $c$ represents the penalty paid by communicative players, and $B_0$ and $\beta_0$ are the maximum benefits paid with joint $CC$ communicators and mixed $CN$ players, respectively. The phase-dependent function $f(\Delta\phi)$ encodes sinusoidal-dependence on the phase-mismatch of the standard Kuramoto model (shifted up to ensure it's always positive). Finally, the benefit asymmetry $\alpha \in [0, 1]$ breaks the symmetry between the payoff for the communicator and the

non-communicator when exactly one player is communicative. We note that the midpoint $\alpha$ = 1/2 corresponds to the symmetric case. Therefore, we have added a factor of 2 to both $2\alpha$ and $2(1-\alpha)$, ensuring that the $\alpha$ = 1/2 reduces to simply $\beta_0 f(\Delta\phi)$, matching previous work [5].

## 4.2 Population setup

Given $N$ players, we associate a pair of weighted, directed graphs to the population. On these graphs, the $N$ nodes represent players and the weighted, directed edges represent games between players. We implement two, related graphs: an "interaction" graph representing *from whom* a player receives payoffs, and a "reproduction" graph representing *to whom* a player can spread its strategy. For simplicity, our reproduction graphs are identical to the interaction graph with a single self-loop added to each dangling node (*i.e.,* each node with no out neighbors); the well-mixed interaction graph has no dangling nodes, and the *Caenorhabditis elegans* (*C. elegans*) graph has five dangling nodes. These self-loops are necessary in the reproduction graph to ensure that each node has positive indegree as required by the Moran process described in the next section. The only effect of adding these self-loops is on the (originally) dangling nodes: now, if they are selected for reproduction, they mutate with probability $\mu$ and do nothing with probability $1-\mu$. This behaviour, combined with the sparsity of dangling nodes (2%), implies that the reproduction graph's self-loops should have negligible impact on the overall dynamics.

## 4.3 Birth-death Moran process

The population is updated according to a birth-death Moran process with exponential fitness [32]. On each turn, the following steps are performed. First, each edge in the interaction graph corresponds to a game between head node $j$ and tail node $i$, the edge's payoff $\pi_{ij}$ is scaled by the edge weight $w_{ij}$, and the relevant payoff $w_{ij}\pi_{ij}$ is awarded to the head node $i$ only. Since the *Caenorhabditis elegans* (*C. elegans*) edge weights are integers, we can also interpret each edge weight as the number of games played between the two nodes. Fig 1B shows an illustration of this process for a single pair of interacting players connected by a pair of weighted, directed edges. The total fitness for node $i$ is the exponential of the product between the selection strength $\delta$ and the sum of payoffs to node $i$, or $f_i = \exp(\delta \sum_j w_{ij}\pi_{ij})$ with the sum over all edges inwardly incident to node $i$. Then, a single focal node is chosen for reproduction with probability proportional to the node's fitness $f_i$. Finally, a node is chosen for replacement amongst the birth node's out-neighbors with probability proportional to the reproduction graph's edge weight. With mutation probability $\mu$, the death node is replaced by a player with a uniformly random strategy; otherwise, it is replaced by a player with the same strategy as the birth node. This birth-death process is repeated for each turn.

## 4.4 Communication frequency

We define the frequency of communicative strategies $f_{comm}(t)$ as the fraction of players employing a communicative strategy $C$ at a given time step. We also define the time-averaged communicative frequency $f_{comm}$ by averaging $f_{comm}(t)$ over the entire simulation. For simulation times long compared to the mutation turnover time $T_{turn} \gg 1/\mu$, the initial, random distribution of strategies will be negligible and the time-average will correspond to the long-time limit. In section 2 of the S1 Text in S1 Appenidx, we derive an analytic expression (Eq. 10 of S1 Text in S1 Appenidx) for $f_{comm}$ in the well-mixed case by incorporating the benefit asymmetry $\alpha$.

## 4.5 Game type nomenclature

Every edge of the interaction graph defines a game between the two players it connects. Using their relative phase difference $\Delta\phi$, we can calculate the payoff matrix. By comparing the order of each of the four entries, we determine the game type using a topological taxonomy [28]. Using this taxonomy, we calculate the ordinal rank of the four entries in the

Eq. 1 payoff matrix and assign a unique name (*e.g.,* dilemma, deadlock, chicken, *etc.*) to each strict, symmetric game type. However, Fig 2B shows that some mixed *CN* games lie on the border between two game types, such as when $\alpha = 1$. The taxonomy [28] classifies these non-strict games according the number and location of ties ("high tie", "middle tie", "double tie", *etc.*). It also defines a convention for choosing one of the neighboring game types to get a binomial nomenclature (*e.g.,* "high harmony", "mid compromise", "double coordination", *etc.*). We follow the same convention for choosing a neighboring game but drop the tie-indicator to keep our figure legends simple. Specifically, referring to Fig 2B, the $\alpha = 1$ tie between deadlock and compromise games formally corresponds to "low [dead]lock", but we label it as deadlock; similarly, the tie between assurance and staghunt is "mid [stag]hunt", but we denote it as staghunt, while the tie between harmony and peace is "mid harmony", simplified to harmony. Finally, the *NN* game type is always "neutral" while the *CC* game type is always "double cooperation", which we denote as just "cooperation".

## 4.6 Plurality game type

At each time step, we calculate the game type for each edge of the interaction graph. We determine the game type by creating a two-by-two payoff matrix of possibilities where both players have the hypothetical option of switching to the other player's strategy/phase pairing. Using the taxonomy discussed in Section 4.5, we then assign a game type to that interaction. We then identify the plurality game type across all player interactions, where each edge's count is weighted by its edge weight. Then, we calculate the frequency of this "plurality game" across all time steps of a given simulation to determine the distribution of games commonly played. This "plurality game-type" metric, as depicted in Fig 7A and Fig 7A, is often dominated by neutral games (between *NN* players) and coordination games (between *CC*) players. To isolate the other game types involved, we also defined a "plurality *mixed* game-type" by only counting games between mixed *CN* or *NC* pairs in the plurality (or labelling the time step as "all-communicative"/"all-noncommunicative", as necessary). This "plurality mixed game-type" shows more variety and is depicted in all of the other figures (Fig 3, Fig 4, Fig 7C and Fig 7C). Note that an edge's game type is dependent on the players' dynmically-evolving communication strategies (N or C) and relative phase $\Delta\phi$, as well as the fixed game parameters $c$, $B_0$, $\beta_0$, and $\alpha$. Furthermore, this metric is only sensitive to the plurality game and therefore provides no information on the presence/absence of minority game types.

## 4.7 Order parameter

Given that the Kuramoto system of coupled oscillators inspired this evolutionary game model, we also define the standard Kuramoto order parameter:

$$\rho = \frac{1}{N} \left| \sum_{j=1}^{N} e^{i\phi_j} \right|$$

(4)

This parameter ranges from zero to one, inclusive, and represents how coherent the population is, with $\rho = 1$ for fully coherent and $\rho = 0$ fully disordered.

## 4.8 Chimera-like index and metastability index

To compare with a previous analysis [24] of chimera-like states *Caenorhabditis elegans* (*C. elegans*) models, we define a pair of indices related to chimera-like quality and metastability [17]. First, we organize the game's nodes into *M* disjoint communities. We split the nodes into communities using the constant Potts model with a Leiden algorithm implemented via the igraph library [33]. In particular, we used a constant Potts model with resolution 0.1, refinement-randomness $\beta = 0.01$, and 2 iterations of the Leiden algorithm. This resulted in $M = 6$ communities $C_m$.

With these disjoint communities $C_m$, we then calculate the time-dependent, community-wise order parameter $\rho_m(t)$ as

$$\rho_m(t) := \frac{1}{N_m} \left| \sum_{j \in C_m} e^{i\phi_j} \right|$$

(5)

across members $j$ of community $C_m$ with size $N_m$. Then, we define a chimera-like index $\chi$

$$\chi = \langle \sigma_{\text{chi}} \rangle_T$$

(6)

where

$$\sigma_{\text{chi}} := \frac{1}{M-1} \sum_{m=1}^{M} \left( \rho_m(t) - \langle \rho_m(t) \rangle_M \right)^2$$

and a metastability index $\lambda$

$$\lambda = \langle \sigma_{\text{met}} \rangle_M$$

(7)

where

$$\sigma_{\text{met}} := \frac{1}{T-1} \sum_{t=1}^{T} \left( \rho_m(t) - \langle \rho_m(t) \rangle_T \right)^2$$

across the $M$ communities and $T$ time steps. The chimera-like index measures the difference in coherence between communities: complete homogeneity between communities (*e.g.,* all fully synchronized *or* fully disordered) corresponds to $\chi = 0$, while having $M$ communities with half fully synchronized ($\rho_m = 1$) and the other half fully disordered ($\rho_m = 0$) for all times yields a maximum $\chi = M/[4(M-1)] = 3/10$ for our $M = 6$ [17]. Likewise, the metastability index $\lambda$ measures how metastable the system is (*i.e.,* transiting between synchronicity and disorder). A system that is fully synchronized or disordered gives $\lambda = 0$; $\lambda$ is maximized for a system spending equal times synchronized and disordered where the variance of the discrete uniform distribution between $\rho_m = 0$ and $\rho_m = 1$ gives $\lambda = 1/4 = 0.25$.

### 4.9 Sensitivity analysis

The simulations use a set of parameters to explore a particular portion of the system's parameter space. Fig 9 extends this analysis to a sensitivity analysis across different rays of the parameter space. Each of the subplots depicts the variation in chimera-like index $\chi$ when changing a single parameter, shown in the subplot title. The values of this parameter are specified in each subplot's legend, with the value used in the main text depicted by a black line, smaller values in blue, and larger values in red. Additionally, each plot is shown as a function of the asymmetry $\alpha$, and vertical error bars depict the standard deviation of the chimera-like index $\chi$ across ten seeds. The subplots depict the sensitivity of the chimera-like index $\chi$ to changes in (A) maximum mixed benefit to maximum joint benefit ratio $\beta_0/B_0$, (B) maximum joint benefit to cost ratio $B_0/c$, (C) number of phases $m$, (D) selection strength $\delta$, and (E) simulation run time $T$.

We note that the chimera-like index is relatively insensitive to changes in most of the parameters depicted in Fig 9. The variations with respect to changing parameter in (A) $\beta_0$, (B) $B_0$ and (C) number of phases $m$ are all less than their standard deviation, showing they are not statistically significant. This insensitivity to the number of phases $m$ shows that the finite-phase approximation of the continuous phase $\Delta\phi$ has converged for the main-text simulations with $m = 20$. Likewise,

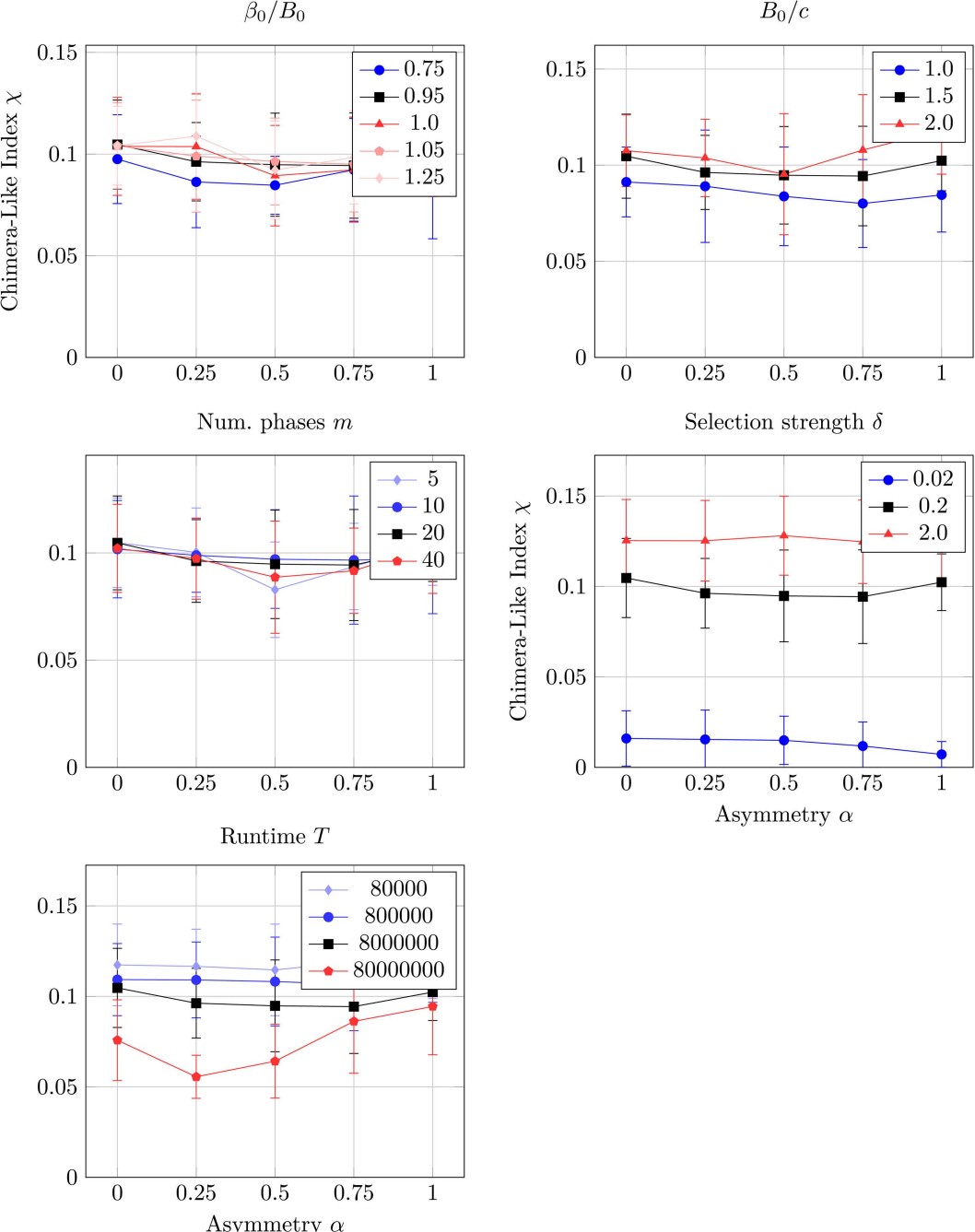

**Fig 9. Sensitivity analysis.** Sensitivity analysis showing how the chimera-like index $\chi$ (Eq. 6) varies when the systems parameters are changed. Each is plotted as a function of the asymmetry $\alpha$. The varied parameters are the (A) maximum mixed benefit to maximum joint benefit ratio $\beta_0/B_0$, **(B)** maximum joint benefit to cost ratio $B_0/c$, **(C)** number of phases $m$, **(D)** selection strength $\delta$, and **(E)** simulation run time $T$. Each subplots' legend shows the values of the varied parameter; the value used in the main text always corresponds to a black line, with smaller value using blue-colored lines and larger values displaying red-colored lines. The vertical error bars depict the standard deviation across ten seeds.

variations in the (E) runtime $T$ have overlapping run times for all but $\alpha = 0.25$ in the longest time $T = 8 \times 10^7$ case, showing that the $T = 8 \times 10^6$ used in the main text is representative of the asymptotic dynamics where transients have died out.

On the contrary, the chimera-like index is very sensitive to the (D) selection strength $\delta$. Interestingly, increasing the selection strength further increases the chimera-like index.

Furthermore, as an additional check on the applicability of approximating the continuous phase difference $\Delta\phi$ with $m$ discrete values, Fig 10 shows the communicative fraction $f_{comm}$ as a function of the maximum joint benefit $B_0$. The vertical error bars depict the standard deviation across ten seeds; their overlap shows that the communicative fraction has also converged for these values of $m$. Therefore, the value of $m = 20$ used in the main text is a good approximation of the continuous phase $\Delta\phi$.

### 4.10 Analytic assumptions

The communicative frequency for the well-mixed population in Fig 3A compared the simulation results to the analytic approximation. The full derivation of this analytic result is presented in Eq. 10 in S1 Text in S1 Appenidx, but here we will highlight the assumptions made in these approximations. The derivation of $f_{comm}$ is only directly applicable to complete graphs, as used in the well-mixed case. Additionally, we assume a small mutation rate $\mu \ll 1$ such that the previous mutation either fixates or goes extinct before the next mutation occurs. As seen in the time-series plots (B–D) of Fig 3 with $\mu = 1 \times 10^{-4}$, the mutations fixate or go extinct almost immediately (less than the plotting sample frequency of 800 time-steps), well before the average mutation time of $1/\mu = 1 \times 10^4$. Thus, the two main assumptions (complete graph and $\mu \ll 1$) of the derivation are satisfied for our choices of setup and parameters.

### Supporting information

**S1 Appendix. Appendix with game graphs and derivations.** Appendix with two-player game order graphs (section 1), derivation of well-mixed communicative fraction with symmetry breaking (section 2), analysis of selection strength $\delta$ influence on $f_{comm}$ (section 3), and analysis of player strategy *vs.* phase influence on chimera-like index $\chi$ (section 4). (PDF)

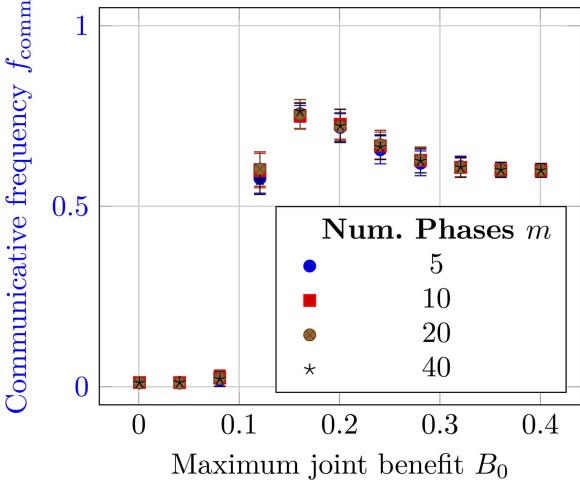

**Fig 10. Sensitivity analysis showing how the communicative fraction $f_{comm}$ varies when the number of phases $m$ is changed.** The legend shows the values of $m$. The vertical error bars depict the standard deviation across ten seeds.

**S1 Video. *C. elegans* $\alpha$ = 0 time evolution.** Time-evolution of *Caenorhabditis elegans* (*C. elegans*) player strategies using the same color scheme as Fig 4D with $B_0/c = 1.5$, $\beta_0/B_0 = 0.95$, $c = 0.1$, $\mu = 0.0001$, $m = 20$, $\delta = 0.2$, $8 \times 10^6$ time steps, and $\alpha = 0$.
(MP4)

**S2 Video. *C. elegans* $\alpha$ = 0.75 time evolution.** Time-evolution of *Caenorhabditis elegans* (*C. elegans*) player strategies using the same color scheme as Fig 4D with $B_0/c = 1.5$, $\beta_0/B_0 = 0.95$, $c = 0.1$, $\mu = 0.0001$, $m = 20$, $\delta = 0.2$, $8 \times 10^6$ time steps, and $\alpha = 0.75$.
(MP4)

**S3 Video. *C. elegans* $\alpha$ = 1 time evolution.** Time-evolution of *Caenorhabditis elegans* (*C. elegans*) player strategies using the same color scheme as Fig 4D with $B_0/c = 1.5$, $\beta_0/B_0 = 0.95$, $c = 0.1$, $\mu = 0.0001$, $m = 20$, $\delta = 0.2$, $8 \times 10^6$ time steps, and $\alpha = 1$.
(MP4)

## Acknowledgments

We thank Dan Rockmore and the Neukom Institute for their cluster-computing support.

## Author contributions

**Conceptualization:** Thomas Zdyrski, Feng Fu.

**Formal analysis:** Thomas Zdyrski.

**Funding acquisition:** Feng Fu.

**Investigation:** Thomas Zdyrski.

**Methodology:** Thomas Zdyrski.

**Project administration:** Feng Fu.

**Software:** Thomas Zdyrski.

**Supervision:** Scott Pauls, Feng Fu.

**Validation:** Feng Fu.

**Visualization:** Thomas Zdyrski.

**Writing – original draft:** Thomas Zdyrski.

**Writing – review & editing:** Scott Pauls, Feng Fu.

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
