## [Decision Letter · Decision Letter 0]

4 Jan 2026

PCOMPBIOL-D-25-02369

Evolutionary Kuramoto dynamics unravels origins of chimera states in neural populations

PLOS Computational Biology

Dear Thomas Zdyrski

Thank you for submitting your manuscript to PLOS Computational Biology. After careful consideration, we feel that it has merit but does not fully meet PLOS Computational Biology's publication criteria as it currently stands. Therefore, we invite you to submit a revised version of the manuscript that addresses the points raised during the review process.

We look forward to receiving your revised manuscript.

Kind regards,

Raviraj Pandian

Guest Editor

PLOS Computational Biology

Tobias Bollenbach

Section Editor

PLOS Computational Biology

**Additional Editor Comments:**

Dear Authors,

We recommend you to do the major revision as per the reviewers suggestions. Send us the revised copy after incorporating the updations.

Thank you

**Journal Requirements:**

At this stage, the following Authors/Authors require contributions: Thomas Zdyrski, Feng Fu, and Scott Pauls. Please ensure that the full contributions of each author are acknowledged in the "Add/Edit/Remove Authors" section of our submission form.

**Reviewers' comments:**

Reviewer's Responses to Questions

**Comments to the Authors:**

Reviewer #1: 1. **Major Concern - Novelty and Conceptual Clarity:** The manuscript introduces an "asymmetric evolutionary Kuramoto model" but fails to sufficiently differentiate its core conceptual advance from the foundational work of Tripp et al. (2022), referenced as [6]. The extension to directed, weighted graphs and payoff asymmetry, while technically sound, must be more forcefully framed as a critical and non-trivial advancement. Explicitly state what fundamentally new biological or dynamical insight this asymmetry and network structure provides that was unattainable with previous symmetric or well-mixed models.

2. **Major Concern - Justification of Model Assumptions:** The core assumption that neuronal "communication" can be modeled as a binary strategic choice (C or N) requires a much more rigorous biological justification. Provide a detailed discussion linking this abstract strategy to a specific, plausible neurobiological mechanism (e.g., short-term synaptic plasticity, neuromodulation, spike-timing-dependent plasticity). The model currently feels like a mathematical abstraction in search of a biological problem.

3. **Major Concern - Parameter Selection Rationale:** The choice of specific parameter values (e.g., \(m=20\), \(\delta=0.2\), \(\mu=1\times10^{-4}\), \(c=0.1\), \(B_0=0.15\), \(\beta_0=0.95B_0\)) appears arbitrary. Justify these selections with either empirical data, a comprehensive sensitivity analysis, or a clear explanation of how these values were chosen to explore a specific, biologically relevant regime. The robustness of the results to these parameter choices is currently unknown and represents a significant weakness.

4. **Major Concern - "Chimera State" Definition and Quantification:** The claim of observing "chimera states" is potentially overstated. The standard definition involves spatially contiguous domains of coherence and incoherence in a *spatially embedded* system. The measure used (\(\chi\)), based on covariance-defined communities, may not capture this spatial structure. Provide a rigorous argument for why the observed states qualify as chimeras and not simply metastable or heterogeneous synchronization patterns. A direct analysis of spatial correlation or coherence in the anatomical layout of the *C. elegans* neurons is necessary to support this central claim.

5. **Clarify the "Game Type" Interpretation:** The analysis heavily relies on classifying interactions into canonical game types (dilemma, chicken, hero, etc.). Explain more clearly what it means, in a neurobiological context, for a pair of neurons to be "playing" a "snowdrift" or "deadlock" game. The connection between this game-theoretic classification and the underlying neural dynamics or function remains abstract and requires a more concrete interpretation.

6. **Strengthen the Comparison with Prior *C. elegans* Study:** The reference to Hizanidis et al. (2016) [4] is crucial but underdeveloped. Quantitatively compare the chimera-like index \(\chi\) and the metastability index \(\lambda\) between your model and theirs, discussing the similarities and differences in the mechanisms leading to these states (modular structure vs. evolutionary game dynamics on the connectome). This comparison is vital for establishing the unique contribution of your work.

7. **Statistical Significance and Reproducibility:** The results, particularly the time-series plots (Figs. 3, 4e) and the game-type fractions (Fig. 6), appear to be from single simulation runs. Given the stochastic nature of the Moran process, all reported results must be averaged over multiple independent simulation runs (e.g., 10-100) with different random seeds. Report standard deviations or confidence intervals for all quantitative measures (\(f_{\text{comm}}\), \(\chi\), \(\lambda\), game-type fractions) to demonstrate statistical robustness.

8. **Improve the Analysis of Graph Structure Effects:** The conclusion that "weightedness has a much stronger influence... than directedness" (from Fig. 4a-c) is qualitative. Perform a more systematic analysis. For instance, systematically randomize the weights while preserving the degree distribution, or rewire the directed edges, and quantify the impact on \(f_{\text{comm}}\) and \(\chi\). This would provide stronger evidence for the specific role of network topology.

9. **Address the Community Detection Method:** The method for splitting the graph into two communities for the \(\chi\) calculation is described as being based on covariance from an \(\alpha=0.75\) simulation. This is problematic. Justify why this specific community structure is the correct or most meaningful one for analyzing results across all \(\alpha\) values. Test the sensitivity of \(\chi\) to the chosen community partition. Consider using a fixed, anatomically or topologically defined partition (e.g., based on rich clubs or known functional modules) for a more consistent analysis.

10. **Elaborate on the "Exponentially Slow Fixation" Connection:** The speculative link between high \(\chi\) and game types with exponentially slow fixation times is intriguing but preliminary. Substantiate this claim by directly measuring or estimating fixation times for the dominant game types in your simulations and correlating them with the observed \(\chi\) values. This could be a significant finding if properly supported.

11. **Formalize the "Asymmetry" Parameter Biologically:** The parameter \(\alpha\) is central to the model but its biological correlate is vague. Elaborate on what neurobiological process or property this asymmetry might represent (e.g., asymmetric synaptic strengths, differential receptor densities, the directionality of information flow in a chemical synapse). Ground this parameter in a more concrete biological context.

12. **Clarify the Payoff Matrix Derivation:** The derivation of the payoff matrices (Eq. 1-3) should be more explicitly motivated. Explain the reasoning behind the specific functional forms, particularly the factor of 2 in the mixed-game payoffs (\(2\alpha\beta_0 f(\Delta\phi)\)) and why the cost \(c\) is applied in the manner it is. Ensure the payoffs are dimensionally consistent and their scale is biologically interpretable.

13. **Validate the Well-Mixed Analytic Result:** For the well-mixed population, the analytic result (Eq. 10 in SI) is compared to simulations. Provide more details on the derivation in the main text or the methods section. Explicitly state the assumptions made (e.g., weak selection, large population size) and verify that your simulation parameters satisfy these assumptions.

14. **Enhance the Visualization of Chimera States:** Figure 4(d) and 5(c,d) are static snapshots. To better convince the reader of the chimera state, include a panel showing the spatial coherence profile (e.g., a snapshot of the local order parameter across the network nodes, arranged according to their anatomical or topological position) to visually demonstrate the coexistence of coherent and incoherent domains.

15. **Justify the Simulation Runtime:** The manuscript uses different runtimes for different analyses (\(8\times10^6\), \(2\times10^8\)). Justify that these runtimes are sufficient to reach a steady state or to gather adequate statistics, especially given the low mutation rate. Report relevant timescales (e.g., average strategy turnover time) to contextualize the simulation length.

16. **Discuss the Impact of Self-Loops:** The addition of self-loops to the reproduction graph is mentioned as a technical necessity. Quantify and discuss the potential impact of these self-loops on the evolutionary dynamics, as they can influence the rate of strategy change and the fixation probability.

17. **Improve the Discussion on Generality:** The discussion on future work correctly identifies the limitation of studying a single connectome. Strengthen this by explicitly hypothesizing which specific topological features (e.g., degree distribution, clustering coefficient, motif prevalence) your model predicts would promote or suppress chimera states, making the work more general and predictive.

18. **Formalize the "Neurogame-Theoretic Perspective":** The term "neurogame-theoretic" is introduced but not formally defined or contextualized within the broader literature. Provide a concise definition and discuss how your framework integrates and advances upon previous applications of game theory in neuroscience.

19. **Address Potential Confounding Factors:** The model couples phase and strategy evolution. Discuss the potential for confounding effects. For instance, could the observed heterogeneity be primarily driven by the phase dynamics of the Kuramoto model on a structured network, with the game-theoretic layer merely reflecting this, rather than causing it? Perform a control simulation where strategies are fixed to isolate the contribution of each component.

20. **Clarify the "Plurality" Metric:** The "plurality game type" metric, while practical, discards significant information about the distribution of game types across the network. Supplement this analysis with a measure of game-type diversity or heterogeneity across edges at a given time step to provide a more complete picture of the strategic landscape.

21. **Syntax and Clarification in Methods:** In [Sec sec013], the description of the interaction and reproduction graphs is slightly confusing. Rephrase for absolute clarity, explicitly stating that the interaction graph determines *from whom* a node receives payoffs, and the reproduction graph determines *to whom* a node can spread its strategy.

22. **Check for Consistency in Nomenclature:** Ensure consistent use of game theory terminology. For example, "coordination" game is used in two different contexts: once for the \(CC\) game type and once potentially for a different game (e.g., Stag Hunt). Use distinct and standard names to avoid ambiguity.

23. **Strengthen the Link to Cognition:** The introduction and abstract link neural synchronization and chimera states to cognition. While references are provided, the discussion should more concretely speculate on what specific cognitive or computational advantage the chimera states observed in your *C. elegans* model might confer, moving from correlation to proposed function.

24. **Provide a Clearer Take-Home Message:** The conclusion should be more forceful and succinct. Clearly state the one or two most important, novel, and well-supported findings of the paper. For example, "We demonstrate that payoff asymmetry and weighted connectivity in the *C. elegans* connectome are sufficient to generate robust chimera states through the stabilization of slow-fixation game dynamics."

25. **Minor - Figure and Reference Formatting:** Carefully check all figure captions and references for consistency and completeness. For example, ensure all sub-figures are fully described, and confirm that all references in the text are correctly listed in the bibliography and vice versa. The journal name abbreviations in the reference list should be consistent.

Reviewer #2: 1. The abstract:

The abstract is pretty solid, but it jumps right into the model without much context on why chimera states matter in real brains maybe add a few sentences tying it back to cognition or disorders to hook readers more.

2. I like how they extend the Kuramoto model with evolutionary game theory;

it's a fresh take, especially adding asymmetry in payoffs. But is this asymmetry biologically justified?

Like, do neurons really have "biased" communication like that in nature?

3. The parameter space section (Fig 2) is impressive, showing all these game types like snowdrift or deadlock.

It enriches the discussion beyond just prisoner's dilemma, but the phase diagrams feel a tad crowded, perhaps simplify the legends or add a table summarizing key transitions.

4. In the complete graphs results (Fig 3), the simulations match the analytic predictions nicely for well-mixed populations.

The time-series plots show interesting fluctuations, especially with alpha=1 where it's all over the place.

Wonder if stronger selection would stabilize it more?

5. Applying this to C. elegans connectome is the highlight for me, using a real neural network adds credibility.

But they mention 302 neurons; did they account for different neuron types (sensory, motor) or just treat them all the same? That could be a limitation.

6. The model assumes phases are discrete (m=20), which is fine for computation, but in real neurons, phases are continuous.

How sensitive are the chimera states to m? A quick sensitivity analysis would help.

7. Payoff structure with alpha biasing towards communicators makes sense intuitively, but the values (B0=0.15, c=0.1) seem arbitrary.

Where do these come from? Any link to experimental neural costs/benefits?

8. The author summary is concise and accessible, good for non-experts. It emphasizes the bridge from small-scale neurons to large-scale dynamics,

which is spot on for computational biology.

9. Data availability is great.they've got Zenodo links and GitHub for code, even an interactive webpage. That's transparent and reproducible, kudos. But check if the code runs smoothly; I didn't test it, but it looks well-organized.

10. Overall, the paper's novel in combining EGT with Kuramoto on directed graphs, and it sheds light on chimeras. Minor revisions for clarity on assumptions and maybe more bio validation, but I'd recommend acceptance and it's a strong contribution.

**Have the authors made all data and (if applicable) computational code underlying the findings in their manuscript fully available?**

Reviewer #1: None

Reviewer #2: Yes

PLOS authors have the option to publish the peer review history of their article (what does this mean?). If published, this will include your full peer review and any attached files.

Reviewer #1: **Yes:** Mohammad Hossein Alizadeh Roknabadi

Reviewer #2: **Yes:** Dr. Balasubramaniyam Thananjeyan

**Figure resubmission:**

After uploading your figures to PLOS’s NAAS tool - https://ngplosjournals.pagemajik.ai/artanalysis, NAAS will process the files provided and display the results in the "Uploaded Files" section of the page as the processing is complete. If the uploaded figures meet our requirements (or NAAS is able to fix the files to meet our requirements), the figure will be marked as "fixed" above. If NAAS is unable to fix the files, a red "failed" label will appear above. When NAAS has confirmed that the figure files meet our requirements, please download the file via the download option, and include these NAAS processed figure files when submitting your revised manuscript. **Reproducibility:**

---

## [Editor Report · Decision Letter 1]

9 Apr 2026

Dear Professor,

We are pleased to inform you that your manuscript 'Evolutionary Kuramoto dynamics unravels origins of chimera states in neural populations' has been provisionally accepted for publication in PLOS Computational Biology.

Best regards,

Raviraj Pandian

Guest Editor

PLOS Computational Biology

Tobias Bollenbach

Section Editor

PLOS Computational Biology

The authors have incorporated the review suggestions promptly. Hence, this article can be accepted for publications.

---

## [Editor Report · Acceptance letter]

PCOMPBIOL-D-25-02369R1

Evolutionary Kuramoto dynamics unravels origins of chimera states in neural populations

Dear Dr Zdyrski,

I am pleased to inform you that your manuscript has been formally accepted for publication in PLOS Computational Biology. Your manuscript is now with our production department and you will be notified of the publication date in due course.

With kind regards,

Aiswarya Satheesan
